# TAFRO Syndrome and COVID-19

**DOI:** 10.3390/biomedicines12061287

**Published:** 2024-06-11

**Authors:** Misato Tane, Hideki Kosako, Takashi Sonoki, Hiroki Hosoi

**Affiliations:** 1Department of Hematology/Oncology, Wakayama Medical University, Wakayama 641-8509, Japan; mtane@wakayama-med.ac.jp (M.T.);; 2Department of Hematology, Kinan Hospital, Wakayama 646-8588, Japan; 3Department of Transfusion Medicine, Wakayama Medical University Hospital, Wakayama 641-8510, Japan

**Keywords:** TAFRO syndrome, COVID-19, cytokine storm

## Abstract

TAFRO syndrome is a systemic inflammatory disease characterized by thrombocytopenia and anasarca. It results from hyperinflammation and produces severe cytokine storms. Severe acute respiratory syndrome coronavirus 2, which led to the coronavirus disease 2019 (COVID-19) pandemic, also causes cytokine storms. COVID-19 was reported to be associated with various immune-related manifestations, including multisystem inflammatory syndrome, hemophagocytic syndrome, vasculitis, and immune thrombocytopenia. Although the pathogenesis and complications of COVID-19 have not been fully elucidated, the pathogeneses of excessive immunoreaction after COVID-19 and TAFRO syndrome both involve cytokine storms. Since the COVID-19 pandemic, there have been a few case reports about the onset of TAFRO syndrome after COVID-19 or COVID-19 vaccination. Castleman disease also presents with excessive cytokine production. We reviewed the literature about the association between TAFRO syndrome or Castleman disease and COVID-19 or vaccination against it. While the similarities and differences between the pathogeneses of TAFRO syndrome and COVID-19 have not been investigated previously, the cytokines and genetic factors associated with TAFRO syndrome and COVID-19 were reviewed by examining case reports. Investigation of TAFRO-like manifestations after COVID-19 or vaccination against COVID-19 may contribute to understanding the pathogenesis of TAFRO syndrome.

## 1. Introduction

TAFRO syndrome is a rare systemic inflammatory disease characterized by thrombocytopenia, anasarca (edema, pleural effusion, and ascites), reticulin fibrosis, renal dysfunction, and organomegaly [1,2]. The annual incidence of TAFRO syndrome was estimated to be 0.9–4.9 per million individuals in Japan [3]. Although the etiology of TAFRO syndrome remains unclear, its pathogenesis involves hyperinflammation [4]. Various diagnostic criteria have been proposed for TAFRO syndrome [1,5,6]. Similar to TAFRO syndrome, Castleman disease also presents with excessive cytokine production. While TAFRO syndrome and Castleman disease exhibit distinct clinical symptoms, there are cases where they overlap [2,7]. In patients with human immunodeficiency virus, Kaposi sarcoma immune reconstitution syndrome was reported to have similar clinical symptoms to Castleman disease [8,9].

Multicentric Castleman disease (MCD) is another inflammatory disease, which is characterized by the overexpression of interleukin (IL)-6 [2,10]. MCD typically causes polyclonal lymphadenopathy. In some cases of MCD, an uncontrolled human herpes virus-8 (HHV-8) infection is the etiological driver for the development of MCD [10]. However, the etiology of HHV-8-negative MCD, i.e., idiopathic MCD (iMCD), has not been elucidated. The diagnoses of iMCD and TAFRO syndrome partially overlap. iMCD is confirmed by performing a histological examination. On the other hand, in TAFRO syndrome lymph node biopsies are sometimes difficult to obtain due to the patient having a bleeding tendency or the size of the target lymph nodes [2]. Some studies have reported a clear distinction between iMCD-TAFRO and TAFRO syndrome, while other researchers consider TAFRO syndrome to be encompassed within iMCD [2,6,11].

There are various clinical causes of cytokine storms besides TAFRO syndrome and MCD, one of which is coronavirus disease 2019 (COVID-19) [12]. COVID-19 is caused by severe acute respiratory syndrome coronavirus 2 (SARS-CoV-2). During the COVID-19 pandemic, various extrapulmonary manifestations of COVID-19 were reported [13]. COVID-19 produces a diverse variety of symptoms, ranging from a mild cold to severe pneumonia, cytokine storms, and multiple organ failure [12]. Most severe manifestations are triggered by hyperinflammatory responses. TAFRO syndrome and COVID-19 share similarities in their clinical presentations and pathogeneses and both involve cytokine storms. Although the association between COVID-19 or vaccination against COVID-19 and the development of TAFRO syndrome or MCD has not been fully investigated, there have been several reports regarding the onset of TAFRO syndrome or MCD after COVID-19 or COVID-19 vaccination. Here, we review the relationships between COVID-19 or COVID-19 vaccination and TAFRO syndrome or MCD, mainly from the perspective of cytokine storms.

## 2. Etiologies of TAFRO Syndrome and MCD Focusing on Viral Infection

The triggers for the onset of TAFRO syndrome remain unclear. The factors that cause TAFRO syndrome have been discussed from three perspectives: autoimmune factors, pathogenic infection factors, and genetic mutations, which are the same as the triggers for MCD [4,14,15]. Infectious diseases, such as viral infections, may precipitate cytokine storms, which is the primary pathology of TAFRO syndrome. However, no viral infections have been definitively shown to cause TAFRO syndrome. Within Castleman disease, a subset of cases is associated with HHV-8 infection, which is classified as HHV-8-associated MCD. In HHV-8-associated MCD, HHV-8 replicates in the lymph nodes and transcribes the viral homolog of IL-6 (vIL-6) [10,15]. HHV-8-positive plasmablasts express both human IL-6 and vIL-6. IL-6 induces B-cell and plasma cell maturation, acute inflammation, and vascular endothelial growth factor (VEGF) secretion [15]. VEGF can induce endothelial cells to secrete more IL-6, thereby forming a positive feedback loop for cytokine storms.

It has been speculated that viral infections may also be a cause of HHV-8-negative MCD, which is referred to as iMCD [10]. A study using virome capture sequencing and a vertebrate virus platform showed that no novel viruses were detected in fresh frozen lymph node tissue from 11 HHV-8-negative iMCD patients [16]. Although Epstein–Barr virus (EBV) was detected in some of these patients, quantitative testing for EBV did not detect systemic EBV viremia, resulting in occult EBV infections in the lymph nodes. On the other hand, some studies have suggested that an association exists between EBV and iMCD [17,18,19]. The involvement of EBV in the pathogenesis of MCD remains unclear. Interestingly, EBV is known to be capable of inducing hemophagocytic lymphohistiocytosis (HLH) and cytokine storms [12]. HHV-8 induces MCD through viral replication within lymph nodes, whereas EBV may induce iMCD through an excessive immune response to eliminate viral-infected cells. While previous studies addressing the relationship between EBV and iMCD lacked detailed descriptions of cytokine dynamics, investigating the impact of viral infections, including EBV, on the pathogenesis of MCD would be worthwhile.

As with iMCD, no virus has been confirmed to be associated with the onset of TAFRO syndrome. Two reports have linked EBV to the development of TAFRO syndrome or TAFRO syndrome-like symptoms [20,21]. In these two cases, in situ hybridization of lymph node biopsy samples showed cells that were positive for EBV-encoded RNA. Recently, we reported a case of severe systemic inflammation mimicking TAFRO that arose after COVID-19, as described in detail in Section 5 [22].

## 3. TAFRO Syndrome and Cytokine Storms

Cytokine storms are a hyperinflammatory state caused by excessive cytokine production and immune dysregulation [12]. While the term cytokine storm initially referred to graft-versus-host disease after allogeneic hematopoietic stem cell transplantation, it is now used for various pathological conditions, including sepsis, viral infections, autoinflammatory disease, HLH, MCD, and immune reactions after chimeric antigen receptor (CAR) T-cell therapy [23].

Although some researchers have demonstrated that TAFRO syndrome and iMCD-TAFRO are different disease entities, they share similar pathophysiologies involving cytokine storms [2,4,7]. For instance, elevated levels of interferon (IFN) γ-inducible protein 10 kDa, IL-10, IL-23, chemokine C-X-C motif chemokine ligand (CXCL) 13, IL-6, and VEGF are reported in iMCD-TAFRO, whereas elevated levels of IL-6 and VEGF are seen in TAFRO syndrome [24,25,26,27,28]. The Janus kinase/signal transducers and activators of transcription (JAK/STAT) and phosphoinositide 3-kinase (PI3K)/Akt/mammalian target of rapamycin (mTOR) pathways have been suggested to be involved in the cytokine storms seen in iMCD-TAFRO and TAFRO syndrome [4,28]. Several previous studies have shown that the PI3K/Akt/mTOR pathway is activated in iMCD-TAFRO [29,30,31,32]. It was suggested that type I IFN may act as a driver of mTOR signaling mediated by JAK in iMCD-TAFRO [31]. In fact, mTOR inhibitor treatment has been shown to be effective in patients with tocilizumab-refractory iMCD-TAFRO [29]. In addition, a case of TAFRO syndrome was successfully treated with a JAK inhibitor [33]. Although the initial cause of the activation of these signaling pathways in iMCD-TAFRO and TAFRO syndrome remains unclear, viral infections may trigger disease flares through the production of type I IFN and pro-inflammatory cytokines [11].

Cytokine storms can occur in patients with malignant tumors. Cancer immunotherapies, such as bi-specific T-cell engagers and CAR-T therapy, have recently been increasing. The cytokine storm that occurs following cancer immunotherapy is referred to as cytokine release syndrome [34]. Similar to Castleman disease, cytokine release syndrome in immunotherapy is characterized by elevated IL-6 levels, and tocilizumab is administered to block the IL-6 receptor [35]. Similarities with the cytokine storm associated with COVID-19 have also been reported [34].

## 4. COVID-19 and Cytokine Storms

The COVID-19 pandemic, which was caused by SARS-CoV-2, has affected millions of people worldwide. During the initial stages of the pandemic, a notable surge in severe COVID-19 cases occurred, which was often attributed to cytokine storms. While numerous reviews regarding immune responses to COVID-19 have been published, this review focuses on elucidating the innate nucleic acid-sensing pathways in operation during SARS-CoV-2 infections and the immune evasion mechanisms that the virus has developed to promote viral replication and infection, resulting in cytokine storms.

Coronaviruses, including SARS-CoV-2, are enveloped, single-stranded RNA (ssRNA) viruses and form double-stranded RNA (dsRNA) during the replicative phase [36]. The innate immune system recognizes pathogen-associated molecular patterns, which comprise viral nucleic acids, proteins, lipids, structural components, and other viral intermediates, including ssRNA or dsRNA, through distinct pattern recognition receptors (PRRs). These PRRs include retinoic acid-inducible gene (RIG)-I-like receptors, Toll-like receptors, nucleotide-binding oligomerization domain (NOD)-like receptors, and C-type lectin-like receptors [37,38,39]. In an infected cell, ssRNA and dsRNA are detected by these PRRs in the cytoplasm and endosomes. Then, the PRRs start a signaling cascade that leads to the production of type I and Ⅲ IFNs and pro-inflammatory cytokines [37,38,39].

While PRRs recognize viral RNA, host damage-associated molecules, like mitochondrial DNA, which are released as a consequence of tissue injury, activate DNA-sensing cytosolic receptors. These cytoplasmic receptors include cyclic GMP-AMP synthase (cGAS) and stimulator of IFN genes (STING). After recognizing cytosolic DNA, cGAS is activated, produces cyclic GMP-AMP, and then binds to STING on the endoplasmic reticulum. Subsequently, STING translocates to the Golgi apparatus and recruits and activates the kinases TANK-binding kinase 1 and IkappaB kinase, activating IFN regulatory factor 3 and nuclear factor-κB, leading to the production of type I and III IFNs and pro-inflammatory cytokines. Moreover, IFN-inducible protein 16 is involved in crosstalk between DNA- and RNA-sensing mechanisms, which activate the STING pathway in a cGAS-independent manner [37,38,39].

Like other viruses, SARS-CoV-2 has various evasion mechanisms for avoiding innate immune responses, such as envelope protein, open reading frames 3a and 8, and non-structure protein 3d [38]. In severe COVID-19, decreased production of anti-viral IFNs and increased production of pro-inflammatory cytokines arise as a result of immune evasion, leading to delayed virus elimination and cytokine storms.

In addition to the acute immune response elicited by the virus during the acute phase, SARS-CoV-2 also induces various symptoms through abnormal host immune responses. While the frequency of severe COVID-19 has decreased from the early phase of the pandemic, long COVID (LC), or post-acute sequelae of COVID-19, persists as an important condition [40]. LC is characterized by worsening or newly emerging symptoms after an acute SARS-CoV-2 infection, which may persist for months or years. This heterogeneous condition produces a variety of symptoms, such as neurological, respiratory, cardiac, and systemic problems. Although the pathophysiology of LC has not been fully elucidated, it has been suggested to be associated with immune dysregulation. For example, functional changes in myeloid cells, which are major producers of pro-inflammatory cytokines, have been observed in LC as well as in acute SARS-CoV-2 infections. In addition, persistently elevated levels of inflammatory cytokines, such as IL-1β, IL-6, tumor necrosis factor (TNF), IFN-α, and IFN-γ, are detected in patients with LC [41,42,43]. These findings imply that cytokine storms could arise even after an acute SARS-CoV-2 infection.

Multisystem inflammatory syndrome in children and adults (MIS-C/A) is a rare but fatal hyperinflammatory state which sometimes develops several weeks after SARS-CoV-2 infection [44,45]. MIS-C/A involves various organ disorders such as of the cardiovascular, respiratory, gastrointestinal, dermatologic, hematologic, and neurologic systems. Although the pathogenesis of MIS-C/A is not well known, post-infectious immune dysregulation, including upregulation of type 1 conventional dendritic cells, was reported to be implicated [46]. Cytokine storm is also important in the pathogenesis of MIS-C/A [47,48]. Elevated levels of IL-1β, IL-6, IL-8, IL-10, IL-17, and IFN-γ were reported in the acute phase of MIS-C patients [49,50,51]. Moreover, anti-cytokine therapies targeting IL-1, IL-6, and TNF-α, in addition to corticosteroids and intravenous immunoglobulin, are effective [44,47].

Severe COVID-19, MIS-C/A, and TAFRO syndrome may manifest similar symptoms. Table 1 summarizes the characteristics of each condition [3,24,44,45,52,53,54,55,56,57,58,59,60,61]. Severe COVID-19 tends to occur in elderly individuals and primarily results in respiratory failure. In contrast, MIS-C/A develops more frequently in relatively young individuals and leads to multi-organ failure involving the cardiovascular, respiratory, dermatologic, hematologic, and gastrointestinal systems. TAFRO syndrome occurs in middle-aged or elderly individuals and is characterized by thrombocytopenia, fluid retention, myelofibrosis, renal failure, and organomegaly.

## 5. TAFRO Syndrome and COVID-19

### 5.1. TAFRO Syndrome and COVID-19 Infections

TAFRO syndrome and excessive immunoreaction after severe COVID-19 can induce similar symptoms and pathophysiologies through cytokine storms. A 61-year-old Japanese female was initially reported to have severe systemic inflammation mimicking TAFRO syndrome following COVID-19 [22]. She developed a fever, anasarca, renal failure, organomegaly, and thrombocytopenia three weeks after COVID-19. Although she was initially diagnosed with multisystem inflammatory syndrome in adults (MIS-A) and administered corticosteroids, her fluid retention, renal failure, and elevated C-reactive protein levels were worsening. A bone marrow examination showed reticulin myelofibrosis. Therefore, she was tentatively diagnosed with TAFRO syndrome. MIS-A is a hyperinflammatory state that arises after SARS-CoV-2 infections, leading to multi-organ dysfunction. Although the aforementioned case was similar to MIS-A in that it involved a cytokine release syndrome developing after COVID-19, the typical cardiovascular, dermatology, and gastrointestinal manifestations of MIS-A were not observed, and the patient’s symptoms lasted longer than they usually do in MIS-A. The patient’s symptoms were clinically consistent with those of TAFRO syndrome and improved after treatment based on that given for TAFRO syndrome. In this case, an antigen test for SARS-CoV-2 was negative at the onset of TAFRO syndrome. Thus, cytokine storms induced by preceding COVID-19 may affect the development of conditions like TAFRO syndrome. There is a possibility that some viruses act as triggers for cytokine storms in TAFRO syndrome, and it is important to investigate any viruses involved in infections that arose just prior to the diagnosis of TAFRO syndrome in addition to viruses detected at the time of the diagnosis of TAFRO syndrome.

To examine the associations between TAFRO syndrome or iMCD and COVID-19, a systematic literature search of PubMed and Google Scholar to identify cases published up to 29 February 2024 was conducted (Table 2). In addition to our previous report, another case from the United States has been reported as iMCD-TAFRO following COVID-19. A 16-year-old female developed worsening fatigue, acute kidney injury, ascites, and cytopenia six weeks after COVID-19 [62]. A lymph node biopsy showed Castleman-like changes, and a bone marrow biopsy showed reticulin fibrosis. She met the criteria for iMCD-TAFRO and received treatment with siltuximab and prednisolone.

Severe COVID-19 in a patient with TAFRO syndrome was also reported [63]. A 66-year-old Japanese male taking immunosuppressive agents, including prednisolone, cyclosporine A, and sarilumab for TAFRO syndrome, showed severe COVID-19. The authors stated that hyperinflammation plays a pivotal role in severe COVID-19 and TAFRO syndrome. There is concern that COVID-19 may exacerbate cytokine storms in patients with iMCD and TAFRO syndrome; however, a recent report suggests that iMCD flare-ups were actually infrequent [64]. Since the COVID-19 pandemic, there have only been a few case reports about the onset of TAFRO syndrome or iMCD. Large-scale cohort studies are warranted to determine the frequency of TAFRO syndrome or iMCD during the COVID-19 pandemic.
biomedicines-12-01287-t002_Table 2Table 2Reported cases of TAFRO syndrome or iMCD after COVID-19 and COVID-19 vaccination.CaseAgeSexVaccineTime to Symptom OnsetTreatmentOutcomeReferenceTAFRO syndrome and COVID-19161FBioNTech BNT 162b7 days after COVID-19Corticosteroid, RTX, PE, CsAImproved[22]iMCD and COVID-19216FNone6 weeks after COVID-19Corticosteroid, siltuximabImproved[62]TAFRO syndrome and COVID-19 vaccination342MBioNTech BNT 162b22 weeks after vaccinationCorticosteroid, tocilizumab, RTXDead[65]445MModerna mRNA-12736 days after vaccinationCorticosteroidImproved[66]iMCD and COVID-19 vaccination520MBioNTech BNT 162b218 days after vaccinationCorticosteroid, siltuximabImproved[67]640MModerna mRNA3 months after vaccinationChemotherapyImproved[68]iMCD, idiopathic multicentric Castleman disease; COVID-19, coronavirus disease 2019; RTX, rituximab; PE, plasma exchange; CsA, cyclosporine A.


### 5.2. TAFRO Syndrome and COVID-19 Vaccination

The mRNA vaccines against COVID-19 are the first approved as mRNA products for clinical use [69]. They have prevented severe COVID-19, but are known to induce several autoimmune and inflammatory disorders. Although immune responses induced by mRNA vaccine against COVID-19 are not fully elucidated, type I IFN signaling was demonstrated to be important for the cytotoxic T-cell response induced by BNT162b2 in mouse models [70]. In humans, elevated levels of cytokines such as IL-6, IFN-γ, and CXCL10 were reported after mRNA vaccination [71,72]. There were no mentions of adverse events related to TAFRO syndrome or iMCD in either the package inserts of COVID-19 vaccines or the reports of the clinical trials for COVID-19 vaccines [73,74].

A systematic literature search of the PubMed and Google Scholar databases was also conducted to find cases of TAFRO syndrome or iMCD that arose after COVID-19 vaccination (Table 2). Two cases of TAFRO syndrome that arose after COVID-19 vaccination were identified. A 42-year-old Japanese male developed a fever that lasted for two weeks after receiving the second dose of the BNT162b2 mRNA COVID-19 vaccine [65]. He was diagnosed with COVID-19 and received corticosteroids, tocilizumab, and rituximab. However, he died three months after the onset of his condition. The author stated that COVID-19 vaccines may cause hyperinflammatory states, which can lead to TAFRO syndrome. The other case involved a 45-year-old Japanese male who developed a persistent fever for six days and renal dysfunction after receiving the first dose of the Moderna mRNA-1273 COVID-19 vaccine [66]. He received corticosteroid therapy, and his condition improved one month after his admission. In the latter case report, the author stated that an abnormal autoimmune response to COVID-19 vaccination caused the overproduction of cytokines, leading to TAFRO syndrome-like symptoms.

There have been two reported cases of iMCD after COVID-19 vaccination. Both patients were diagnosed with iMCD based on lymph node biopsies. The first case involved a 20-year-old male from Germany who developed a fever and swollen axillary lymph nodes after receiving the second dose of the BNT162b2 COVID-19 mRNA vaccine [67]. He was diagnosed with iMCD with TAFRO syndrome and received siltuximab, which led to a rapid improvement. The second case reported from Taiwan involved a 40-year-old male who experienced progressive symptoms, including a cough, dyspnea on exertion, and fever, for three months after receiving the first dose of the COVID-19 mRNA vaccine (Moderna) [68]. He was diagnosed with iMCD and received chemotherapy. A previous study showed that mRNA vaccine administration increased IL-6 levels [71]. The elevation of IL-6 levels by vaccination may serve as a trigger for the onset of iMCD. On the other hand, a cohort study showed that only 1 of 112 patients with iMCD developed disease flare [64]. Further studies are needed to elucidate the association of iMCD and vaccination against COVID-19.

Several reports have described the development of MIS-A following COVID-19 vaccination [75,76]. Although the incidence of MIS-A after COVID-19 vaccination is very low, it is a pathological condition that clinicians should be aware of, as it can be fatal [77]. In addition to MIS-A, there are also autoimmune-related inflammatory conditions that are difficult to differentiate from TAFRO syndrome [78,79]. Excessive inflammation sometimes occurs after COVID-19 mRNA vaccination, leading to TAFRO syndrome, iMCD, MIS-A, or autoimmune disorders [80,81,82]. The pathophysiology of these post-vaccination conditions remains largely unclear, and further research, including regarding cytokine dynamics, is warranted to elucidate the pathogenesis of hyperinflammatory states and cytokine storms.

A systematic literature search was conducted using the PubMed and Google Scholar databases to investigate cases of TAFRO syndrome or iMCD following vaccinations other than mRNA vaccines for COVID-19. Within our search scope, no reports were found of TAFRO syndrome or iMCD occurring after the administration of other vaccines, such as the influenza vaccine. Additionally, MIS-C/A is a concept that emerged after the COVID-19 pandemic [83]. Further investigations are needed to assess whether the onset of TAFRO syndrome, iMCD, or MIS-A following vaccination is unique to mRNA vaccines.

## 6. Genetic Predisposition of iMCD, TAFRO Syndrome, and Severe COVID-19

The involvement of genetic factors in cytokine storms has been reported. Pathological genetic mutations, such as in *PRF1*, *UNC13D*, *STXBP2*, *RAB27A*, and *XIAP*, are commonly detected in patients with primary HLH [84]. In patients with secondary HLH triggered by viral or autoimmune disorders, heterozygous polymorphisms are often found in the same genes that exhibit mutations in primary HLH [12].

In patients with iMCD, germline mutation of *FAS* and somatic mutation of *NCOA4* have been reported [85,86]. While reports of mutations in iMCD are expected to accumulate, a review suggested that mutations are commonly found in genes involved with chromatin organization and genes associated with the mitogen-activated protein (MAP) kinase pathways [87]. In iMCD-TAFRO, *DNMT3A* somatic mutation was reported in a sequencing study [88]. In addition, another study showed two major gene mutations, a somatic *MEK* mutation and a germline *RUNX1* mutation [89]. In the latter study, the *MEK* mutation caused hyperactivated MAP kinase signaling and conferred IL-3 hypersensitivity. The author demonstrated that *MEK* and *RUNX1* mutations are pathogenic for iMCD-TAFRO. Since the number of cases of TAFRO syndrome examined for these genetic mutations is limited, further studies are needed to assess the frequency of genetic mutations in iMCD and TAFRO syndrome.

Multiple genetic mutations related to susceptibility to SARS-CoV-2 infection and the severity of COVID-19 are being investigated. It is known that SARS-CoV-2 infects epithelial cells in the nasopharynx via the angiotensin-converting enzyme 2 (ACE2) receptor [90]. Previous genome-wide association studies (GWASs) suggested that a variant upstream of the *ACE2* gene may be associated with susceptibility to SARS-CoV-2 infection [91,92]. In addition, GWASs have been conducted to explore the genetic variants associated with the severity of COVID-19 in addition to susceptibility to SARS-CoV-2 [93]. Several studies indicated that variants in genes that modulate the immune response to viral infections are also associated with COVID-19 disease severity. These genes included *TYK2*, *IFNAR2* (IFNα and IFNβ receptor subunit 2/3), and *OAS1* [94,95,96]. Moreover, specific human leukocyte antigen alleles have been reported to be associated with adverse events, including fever and fatigue after administration of mRNA vaccines against COVID-19 [97,98,99].

Although the same gene mutations associated with the pathophysiology have not been detected among iMCD/TAFRO syndrome, severe COVID-19, and vaccination against COVID-19, different mutations can activate the common signaling pathway to cause cytokine storms. Further investigations on gene expression profiles, in addition to genetic polymorphisms, will be helpful in elucidating the pathophysiology of cytokine storms and related syndromes.

## 7. Conclusions

TAFRO syndrome is a relatively novel disease entity, involving systemic inflammation, which was first reported in 2010. It is a heterogeneous disease, with various triggers presumed to be responsible for its onset. Since March 2020, COVID-19 has caused a pandemic, and various accompanying and complicating symptoms have been reported as numerous patients have been infected. Cytokine storms are the key pathogenetic factor underlying the severity of COVID-19, and it has been reported that COVID-19 can produce symptoms common to diseases characterized by cytokine storms. Cytokine storms are the core pathophysiology of TAFRO syndrome and iMCD, and symptoms similar to those seen in TAFRO syndrome and iMCD were reported to occur after COVID-19 or COVID-19 vaccination. No cohort studies have been published demonstrating an increase in TAFRO syndrome or iMCD during the COVID-19 pandemic. Thus, this review summarizes the reported TAFRO syndrome and iMCD cases after COVID-19 or vaccination against COVID-19. These conditions share the common feature of cytokine storms. Therefore, investigation of TAFRO-like manifestations after COVID-19 or vaccination against COVID-19, mainly focusing on cytokine dynamics and genetic predisposition, will contribute to a better understanding of the pathogenesis of TAFRO syndrome.

## Figures and Tables

**Table 1 biomedicines-12-01287-t001:** Typical characteristics of severe COVID-19, MIS-C/A, and TAFRO syndrome.

Characteristics	Severe COVID-19	MIS-C/A	TAFRO
Clinical manifestation	Respiratory failure, fever, coagulation abnormality	Fever, cardiac dysfunction, myocarditis, multi-organ dysfunction	Fever, thrombocytopenia, anasarca, myelofibrosis, renal failure
Pathophysiology	Cytokine storm triggered by SARS-CoV-2 infection	Delayed dysregulated immune response after COVID-19	Cytokine storm; trigger is not identified
Pathogenic cytokine driver	IL-6, IL-8, IL-1β, TNF-α, MCP-1	IL-6, IL-10	IL-6, VEGF, CXCL10
Age	Elderly individuals	Children, young adults	Middle-aged, elderly individuals
Sex (M:F)	6–7:3–4	7:3	5:5
Treatment	Corticosteroids, tocilizumab, baricitinib, anti-SARS-CoV-2 drugs	Corticosteroids, IVIG, anakinra	Corticosteroids, rituximab, tocilizumab, CsA
Outcome	Mortality rate: 10–20% (initially 60%)	Mortality rate: 0–10%	Five-year survival rate: 67%

COVID-19, coronavirus disease 2019; MIS-C/A, multisystem inflammatory syndrome in children and adults; SARS-CoV-2, severe acute respiratory syndrome coronavirus 2; IL, interleukin; TNF, tumor necrosis factor; MCP, monocyte chemoattractant protein; VEGF, vascular endothelial growth factor; CXCL, C-X-C motif chemokine ligand; IVIG, intravenous immunoglobulin; CsA, cyclosporine A.

## Data Availability

No new data were created in this study. Data sharing is not applicable to this article.

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
