# Peer review of "TAFRO Syndrome and COVID-19"

_biomedicines, 2024, doi:10.3390/biomedicines12061287_

Round 1
Reviewer 1 Report
Comments and Suggestions for Authors
Thank you for submitting this review of TAFRO syndrome and its potential associations with the inflammatory condition related to COVID-19 infections and vaccinations. I have a few minor comments:
1. Table 2 lists the published cases of either TAFRO or iMCD and COVID-19 infections or vaccinations. It would be helpful to have the title of the Table describe it as a cases of both COVID-19 vaccination and infection.
2. In section 5.2 TAFRO syndrome and COVID-19 vaccinations, it would be helpful to know if there were any cases of TAFRO, iMCD, MIS-C/A reported in the literature related to anyother commonly given vaccinations using the same methods (PubMed / Google Scholar).
3. Related to about comment, it would be usefull if you stated and referenced the actual incidence of reported severe adverse events (SAEs) and/or adverse events (AEs) from the clinical studies for the available COVID-19 vaccines (BioNTech and Moderna), which are listed in their package inserts / macufacture fact sheet. The are also detailed in the orginal publications from the trials that were done to achieve approval for the vaccines. Althought the SAE/AE lists will not state TAFRO or iMCD specifically, it will contain rates of some shared symptoms (fever, cytopenias..etc).
Author Response
Point-by-point replies to the comments from reviewer #1
Thank you very much for giving us insightful comments and important advice. We agree with your comments and have incorporated them into the revised version of our manuscript.
Reviewer’s comment
Thank you for submitting this review of TAFRO syndrome and its potential associations with the inflammatory condition related to COVID-19 infections and vaccinations. I have a few minor comments:
Response to reviewer
Thank you for reviewing our manuscript. Our point-by-point responses are shown below.
Comment 1
Table 2 lists the published cases of either TAFRO or iMCD and COVID-19 infections or vaccinations. It would be helpful to have the title of the Table describe it as a cases of both COVID-19 vaccination and infection.
Our response
Thank you for your comments. I have changed the title of Table 2, as you indicated.
Comment 2
In section 5.2 TAFRO syndrome and COVID-19 vaccinations, it would be helpful to know if there were any cases of TAFRO, iMCD, MIS-C/A reported in the literature related to any other commonly given vaccinations using the same methods (PubMed / Google Scholar).
Our response
Thank you for your critical comments. As you pointed out, a similar search was conducted using the terms “TAFRO”, “Castleman”, and “vaccination”. However, no reports were found linking TAFRO syndrome or iMCD to vaccinations other than those for COVID-19. Additionally, MIS-C/A is a concept that emerged after COVID-19, and no reports were found prior to the COVID-19 pandemic. These points have been added in the manuscript. (lines 307-313)
Comment 3
Related to about comment, it would be usefull if you stated and referenced the actual incidence of reported severe adverse events (SAEs) and/or adverse events (AEs) from the clinical studies for the available COVID-19 vaccines (BioNTech and Moderna), which are listed in their package inserts /macufacture fact sheet. The are also detailed in the orginal publications from the trials that were done to achieve approval for the vaccines. Althought the SAE/AE lists will not state TAFRO or iMCD specifically, it will contain rates of some shared symptoms (fever, cytopenias..etc).
Our response
Thank you for your important comment. As you pointed out, we confirmed adverse events for each of the COVID-19 vaccines from BioNTech and Moderna-1273 through their package inserts and the clinical trials leading to their approval. There were no mentions of adverse events related to TAFRO syndrome or iMCD in either the package inserts or reports of the clinical trials. We have added this information to the main text (lines 266-268). We considered your suggestions to include the incidence rates of adverse events such as fever and thrombocytopenia, which are observed in TAFRO syndrome and iMCD. However, since fever and thrombocytopenia can also result from other causes, such as allergic reactions or thrombosis, we decided not to include this information in the main text to avoid confusing the readers.
Thank you for taking the time to review our manuscript.

Reviewer 2 Report
Comments and Suggestions for Authors
The presented manuscript to be condisered for being published as a review is a very interesting manuscript about TAFRO syndrome, which is a systemic inflammatory disease characterized by thrombocytopenia and anasarca. It results from hyperinflammation and produces severe cytokine storms. Severe acute respiratory syndrome coronavirus 2, which led to the coronavirus disease 2019 (COVID-19) pandemic, also causes cytokine storms. COVID-19 was reported to be associated with various immune-related manifestations, including multisystem inflammatory syndrome, hemophagocytic syndrome, vasculitis, and immune thrombocytopenia. In the end of the manuscript, the co-authors of this manuscript suggest that further investigations of TAFRO-like manifestations after COVID-19 or vaccination against COVID-19 may contribute to understanding the pathogenesis of TAFRO syndrome. It is a well written manuscript, with the proper references and updated relative citations. I would suggest to the Editor in Chief to accept it for publication in the present form.
Author Response
Point-by-point replies to the comments from reviewer #2
Reviewer’s comment
The presented manuscript to be condisered for being published as a review is a very interesting manuscript about TAFRO syndrome, which is a systemic inflammatory disease characterized by thrombocytopenia and anasarca. It results from hyperinflammation and produces severe cytokine storms. Severe acute respiratory syndrome coronavirus 2, which led to the coronavirus disease 2019 (COVID-19) pandemic, also causes cytokine storms. COVID-19 was reported to be associated with various immune-related manifestations, including multisystem inflammatory syndrome, hemophagocytic syndrome, vasculitis, and immune thrombocytopenia. In the end of the manuscript, the co-authors of this manuscript suggest that further investigations of TAFRO-like manifestations after COVID-19 or vaccination against COVID-19 may contribute to understanding the pathogenesis of TAFRO syndrome. It is a well written manuscript, with the proper references and updated relative citations. I would suggest to the Editor in Chief to accept it for publication in the present form.
Response to reviewer
Thank you for reviewing our manuscript and giving us your favorable comments.

Reviewer 3 Report
Comments and Suggestions for Authors
The authors did an excellent work on this topic and this paper in my opinion is well written and summarize well majority of the relevant detail on this topic.
In order to improve the paper, I would suggest the following additions:
1) Line 36- it should be added that in HIV infected individuals IRIS and KCIS can present similarly (A Fatal Case of Kaposi Sarcoma Immune Reconstitution Syndrome (KS-IRIS) Complicated by Kaposi Sarcoma Inflammatory Cytokine Syndrome (KICS) or Multicentric Castleman Disease (MCD): A Case Report and Review - PubMed (nih.gov)
AND
Clinical Features and Outcomes of Patients With Symptomatic Kaposi Sarcoma Herpesvirus (KSHV)-associated Inflammation: Prospective Characterization of KSHV Inflammatory Cytokine Syndrome (KICS) - PubMed (nih.gov)
2) Etiology section starting with line 92- is there any data about CMV triggered TAFRO or MCD?
3) Table 1- please add two other columns: KCIS and cytokine release syndrome associated with various immunotherapies for cancer treatment
Cytokine Release Syndrome in the Immunotherapy of Hematological Malignancies: The Biology behind and Possible Clinical Consequences - PMC (nih.gov)
Cytokine release syndrome is becoming an emerging issue with various immunotherapies for cancer patients, and it would be important to highlight the similarities and differences with other etiologies already mentioned above
Comments on the Quality of English Language
minor edits
Author Response
Point-by-point replies to the comments from reviewer #3
Reviewer’s comment
The authors did an excellent work on this topic and this paper in my opinion is well written and summarize well majority of the relevant detail on this topic.
In order to improve the paper, I would suggest the following additions:
Response to reviewer
Thank you for your favorable comments. Our point-by-point responses are shown below.
Comment 1
Line 36- it should be added that in HIV infected individuals IRIS and KCIS can present similarly (A Fatal Case of Kaposi Sarcoma Immune Reconstitution Syndrome (KS-IRIS) Complicated by Kaposi Sarcoma Inflammatory Cytokine Syndrome (KICS) or Multicentric Castleman Disease (MCD): A Case Report and Review - PubMed (nih.gov)
AND
Clinical Features and Outcomes of Patients With Symptomatic Kaposi Sarcoma Herpesvirus (KSHV)-associated Inflammation: Prospective Characterization of KSHV Inflammatory Cytokine Syndrome (KICS) - PubMed (nih.gov)
Our response
Thank you for pointing out the addition of important disease concepts. Additionally, thank you for introducing relevant literature. The following descriptions were added: “In human immunodeficiency virus-infected patients, Kaposi sarcoma immune reconstitution syndrome was reported to have similar clinical symptoms to Castleman disease [8, 9].” (lines 38-40)
Comment 2
Etiology section starting with line 92- is there any data about CMV triggered TAFRO or MCD?
Our response
Thank you for your comments. As you pointed out, we also considered that cytomegalovirus (CMV) could potentially trigger the onset of TAFRO syndrome and conducted a literature search. However, within the scope of our search, we did not find any case reports of TAFRO syndrome or MCD developing after CMV infection.
Comment 3
Table 1- please add two other columns: KCIS and cytokine release syndrome associated with various immunotherapies for cancer treatment
Cytokine Release Syndrome in the Immunotherapy of Hematological Malignancies: The Biology behind and Possible Clinical Consequences - PMC (nih.gov)
Cytokine release syndrome is becoming an emerging issue with various immunotherapies for cancer patients, and it would be important to highlight the similarities and differences with other etiologies already mentioned above
Our response
Thank you for your suggestions. As you pointed out, many other situations also result in cytokine storms, similar to severe COVID-19, MIS-C/A, and TAFRO syndrome. As suggested, cytokine release syndrome can also occur after various immunotherapies for cancer treatment, including CAR-T therapy. As you proposed, we have added a discussion in the text regarding cytokine storms arising after cancer treatment. (lines 125-131) On the other hand, Table 1 illustrates the characteristics of cytokine storms related to COVID-19 and TAFRO syndrome, which is the main theme of this review. As you pointed out, organizing the disease that can cause cytokine storms in the Table is meaningful. However, this paper is not a comprehensive review of cytokine storms, so organizing all conditions that can cause cytokine storms would overlap with the context of other reviews and risk making this review redundant. To avoid reader confusion, we did not include the characteristics of KCIS or cytokine release syndrome after cancer treatment.

Reviewer 4 Report
Comments and Suggestions for Authors
Very interesting paper with novel approach to analysis of cytokine storm in iMCD-TAFRO. Authors emphasis is on similarities of immune response to COVID-19 and immune abnormalities associated with iMCD-TAFRO. That is the strongest point of the paper. Question posed by the reader is if these similarities are due to general structural cytokine response to any inducing force no matter if infectious, inflammatory or other or there is some uniform cause that connects COVID-19, iMCD-TAFRO, i-MDC , HLH and other entities characterized by cytokine storm. Looking at heterogeneity of these entities at the present it seems that former is what has more support in research and clinical practice. That is why author's attempt to connect COVID-19 and COVID-19 vaccination with i-MCD TAFRO seem to be overreaching. Authors need to analyze if incidence of iMDC TAFRO in patients with COVID-19 infection or after COVID-19 vaccination differs from incidence in general population. I would say no. I would say that statement by authors that " Since the COVID-19 epidemics, there have been only a few case reports about the onset of TAFREO syndrome or i-MDC" sound truer. That is why, I think, big part of this paper devoted to few cases of iMCD-TAFRO in CVID-19 patients and after COVID-19 vaccination need to be re-written with analysis of incidence of cases and explanation about shortcomings of cases reports and conclusions based on them.
Specific point:
On page 2/12 to illustrate association between EBV and MCD authors quote review showing that 34% of patients with MCD had a current or previous EBV infection. However, in US the seroprevalence of EBV antibody ranges from 50% in children to 89% in teenagers (Lunn RM et al.,(2017) Philos Trans R Soc Lond B Biol Sci 372).
Author Response
Point-by-point replies to the comments from reviewer #4
Reviewer’s comment
Very interesting paper with novel approach to analysis of cytokine storm in iMCD-TAFRO. Authors emphasis is on similarities of immune response to COVID-19 and immune abnormalities associated with iMCD-TAFRO. That is the strongest point of the paper. Question posed by the reader is if these similarities are due to general structural cytokine response to any inducing force no matter if infectious, inflammatory or other or there is some uniform cause that connects COVID-19, iMCD-TAFRO, i-MDC , HLH and other entities characterized by cytokine storm. Looking at heterogeneity of these entities at the present it seems that former is what has more support in research and clinical practice. That is why author's attempt to connect COVID-19 and COVID-19 vaccination with i-MCD TAFRO seem to be overreaching. Authors need to analyze if incidence of iMDC TAFRO in patients with COVID-19 infection or after COVID-19 vaccination differs from incidence in general population. I would say no. I would say that statement by authors that " Since the COVID-19 epidemics, there have been only a few case reports about the onset of TAFREO syndrome or i-MDC" sound truer. That is why, I think, big part of this paper devoted to few cases of iMCD-TAFRO in CVID-19 patients and after COVID-19 vaccination need to be re-written with analysis of incidence of cases and explanation about shortcomings of cases reports and conclusions based on them.
Response to reviewer
Thank you for your critical comments. As you pointed out, this review focuses on the similarities in pathologies that arise after COVID-19 and in TAFRO syndrome, and it has not been clarified whether COVID-19 or vaccination against COVID-19 directly leads to the manifestation of TAFRO syndrome. The fact that no cohort studies have been published showing an increase in TAFRO syndrome during the COVID-19 pandemic may indicate that cases of TAFRO syndrome triggered by COVID-19 are rare. While future cohort studies are needed to investigate this, there is currently no cohort study data examining the incidence of TAFRO syndrome before and during the COVID-19 pandemic. Therefore, our review is based on case studies focusing on the relationship between TAFRO syndrome, or iMCD, and COVID-19. We added the following description to the conclusion: “No cohort studies have been published demonstrating an increase in TAFRO syndrome or iMCD during the COVID-19 pandemic. Thus, this review summarizes reported TAFRO syndrome and iMCD cases after COVID-19 or vaccination against COVID-19. These conditions share the common feature of cytokine storms.” (lines 362-365)
Comment 1
On page 2/12 to illustrate association between EBV and MCD authors quote review showing that 34% of patients with MCD had a current or previous EBV infection. However, in US the seroprevalence of EBV antibody ranges from 50% in children to 89% in teenagers (Lunn RM et al.,(2017) Philos Trans R Soc Lond B Biol Sci 372).
Our response
Thank you for your comments. The relevant section was a quote from a previous report, but as you pointed out, the content was confusing for readers, so we have removed this sentence. Thank you for pointing this out and providing the reference.

Round 2
Reviewer 4 Report
Comments and Suggestions for Authors
Although paper deals with somewhat controversial correlation between two possibly unrelated entities, I think it is very important to be reviewed and followed closely in the future.